# Structure of lasso peptide epimerase MslH reveals metal-dependent acid/base catalytic mechanism

Yu Nakashima [1,4], Atsushi Kawakami [2,4], Yasushi Ogasawara [3], Masatoshi Maeki [3], Manabu Tokeshi [3], Tohru Dairi [3] ✉ & Hiroyuki Morita [1] ✉

The lasso peptide MS-271 is a ribosomally synthesized and post-translationally modified peptide (RiPP) consisting of 21 amino acids with D-tryptophan at the C-terminus, and is derived from the precursor peptide MslA. MslH, encoded in the MS-271 biosynthetic gene cluster (msl), catalyzes the epimerization at the Cα center of the MslA C-terminal Trp21, leading to epi-MslA. The detailed catalytic process, including the catalytic site and cofactors, has remained enigmatic. Herein, based on X-ray crystallographic studies in association with MslA core peptide analogues, we show that MslH is a metallo-dependent peptide epimerase with a calcineurin-like fold. The crystal structure analysis, followed by site-directed mutagenesis, docking simulation, and ICP-MS studies demonstrate that MslH employs acid/base chemistry to facilitate the reversible epimerization of the C-terminal Trp21 of MslA, by utilizing two pairs of His/Asp catalytic residues that are electrostatically tethered to a six-coordination motif with a Ca(II) ion via water molecules.

Ribosomally synthesized and post-translationally modified peptides (RiPPs), such as lasso peptides with a characteristic N-terminal peptide chain-derived macrocyclic ring structure, are a large, structurally diverse group produced by various organisms, including eukaryotes, eubacteria, and archaea[1–3]. Wide varieties of post-translational modifications, such as epimerization, cyclization, adenylation, glycosylation, and prenylation, of these ribosomally synthesized simple peptides have been reported in RiPP biosynthesis[4–6], and are recognized as crucial steps leading to their structural diversity and distinct bioactivities, such as antibacterial, antiviral, and anticancer properties[6–8]. Furthermore, research to produce various non-natural RiPPs has attracted keen attention, due to their high bioengineering tolerance[9–11]. From this perspective, it is crucial to unravel their complex biosynthetic pathways, especially in the modification steps. One of the current hot topics is the biosynthetic mechanism of D-amino acid-containing RiPPs[4–6], since the D-amino acid-containing molecules

generally serve vital functions in primary and secondary metabolism as antibiotics, hepatotoxins, and hormones[12,13]. Currently, three mechanistically distinct, major systems of L-amino acid to D-amino acid epimerizations with radical S-adenosylmethionine-dependent enzymes, exemplified by PoyD[14] and YydG[15], α/β-hydrolase family enzymes like BotH[16] and LinHL[17,18], and a recently discovered mechanistically mysterious epimerase, MslH[19], have been elucidated in RiPP biosynthesis.

The mslH gene was discovered in the biosynthetic gene cluster of the lasso peptide MS-271 (msl) with a calmodulin-activated myosin light-chain kinase inhibitory activity in Streptomyces sp. M-271[20,21], together with mslA (a ribosomally synthesized precursor peptide consisting of a 21 amino acid N-terminal leader peptide and a 21 amino acid C-terminal core peptide), mslB1 (precursor peptide recognition element), mslB2 (leader peptide excision protease), mslC (macrolactam synthetase), and mslE and mslF (disulfide oxidoreductases)[19,22].

[1]Institute of Natural Medicine, University of Toyama, 2630-Sugitani, Toyama 930-0194, Japan. [2]Graduate School of Chemical Sciences and Engineering, Hokkaido University, N13-W8, Kita-ku, Sapporo, Hokkaido 060-8628, Japan. [3]Graduate School of Engineering, Hokkaido University, N13-W8, Kita-ku, Sapporo, Hokkaido 060-8628, Japan. [4]These authors contributed equally: Yu Nakashima, Atsushi Kawakami. ✉e-mail: dairi@eng.hokudai.ac.jp; hmorita@inm.u-toyama.ac.jp

Previous research using *Streptomyces griseorubiginosus*, another MS-271 producer, demonstrated that its MslH (86% identity with MslH in *Streptomyces* sp. M-271) converted the *C*-terminal L-Trp21 residue on the ribosomally synthesized precursor peptide, MslA, whose leader peptide forms a complex with MslB1, to the D-form (*epi*-MslA) without requiring any cofactors or metals, before the leader peptide excision by MslB2, macro-cyclization reactions by MslC, and disulfide bond formation (by MslE and MslF) occurred (Fig. 1)[19]. This previous study also suggested that MslH catalyzed a reversible epimerization reaction in an in vitro reaction. Based on the previous Phyre2 analysis[19], MslH was suggested to form a calcineurin-like fold, a typical conformation classically found in phosphoesterase domains in phosphodiesterases, protein (serine/threonine) phosphatases, and 5'-nucleotidases, known as metal-dependent enzymes[23–25]. The amino acid sequence alignment demonstrated that several amino acid residues involved in metal coordination in metallo-dependent phosphatases are conserved in MslH[19], suggesting that MslH would also be a metal-dependent enzyme, although epimerases with calcineurin-like folds have never been reported. Furthermore, the detailed catalytic mechanism of MslH, including the metal(s) and the residues responsible for the epimerization of the *C*-terminal L-Trp21 of the substrate, MslA, has remained unknown.

Herein, we report the catalytic mechanism of MslH, a promising epimerase for RiPPs, based on its crystal structure in complex with MslA analogues. The crystal structure analysis, followed by site-directed mutagenesis, ICP-MS studies, and substrate docking studies, unveiled not only the substrate-binding site of MslH, but also the metal-dependent catalytic system of MslH, employing two pairs of His/Asp catalytic dyads electrostatically linked via water molecules to a six-coordination motif with a Ca(II) ion.

## Results

### Overall structure of MslH

To investigate the catalytic mechanism of MslH, we performed crystallographic studies. Recombinant *N*-terminally His₆-tagged MslH was expressed in *Escherichia coli* and purified for crystallization. The purified recombinant MslH migrated as a single band with a molecular weight of 48.5 kDa on SDS−PAGE, in good agreement with the calculated value of 49.4 kDa (Supplementary Fig. 1a). In contrast, a gel-filtration experiment indicated a molecular weight of 410 kDa, suggesting that the recombinant MslH is an octameric enzyme (Supplementary Fig. 2). The X-ray crystal structure of MslH:apo, indexed in the I422 space group with one monomer in the asymmetric unit, was solved at 2.30 Å resolution. The MslH:apo structure was initially determined by the sulfur single-wavelength anomalous diffraction (S-SAD)

method, since the molecular replacement (MR) attempts using the templates suggested by the Phyre2 analysis[19] as search models were unsuccessful (Supplementary Fig. 3a and Table 1). Consequently, a recombinant *N*-terminally His₆-tagged full-length MslA was prepared by co-expression with MslB1 in *E. coli*, since MslA expressed in *E. coli* was only soluble when complexed with MslB1[19]. However, the complex structure with MslH and full-length MslA could not be obtained by co-crystallization and soaking experiments, perhaps due to the reaction on wild-type MslA. Hence, recombinant *N*-terminally His₆-tagged MslAΔTrp21, which lacked the *C*-terminal Trp21, and MslA Trp21G variants, with an achiral glycine instead of Trp21, were prepared by co-expression with MslB1 in *E. coli*. The co-crystallization was accomplished by mixing MslH:MslAΔTrp21-MslB1 or MslH:MslA Trp21G-MslB1 in a 1:1.2 mole ratio. As a result, the X-ray crystal structures of the MslH:MslAΔTrp21 and MslH:MslA Trp21G complexes at 2.29 Å and 2.12 Å resolutions, respectively, with one monomer in the asymmetric unit, which are the same as the MslH:apo structure, were successfully obtained (Supplementary Table 1). Both crystal structures were solved by the MR method, using the MslH:apo structure as the search model (Fig. 2), and the enzyme formed a similar biologically active octamer to that of the MslH:apo structure (Fig. 3 and Supplementary Fig. 4). The MslH structures complexed with MslA analogues adopted almost the same conformations as the MslH:apo structure, with Cα Root Mean Square Deviations (RMSDs) of 0.17 Å (MslH:apo vs. MslH:MslAΔTrp21) and 0.16 Å (MslH:apo vs. MslH:MslA Trp21G) (Supplementary Fig. 3b, c).

Each monomer is composed of a large domain with a grip-like subdomain (G-domain: Ala141-Pro234) and a cover-like subdomain (C-domain: Ser317-Trp363) (Fig. 3a and Supplementary Figs. 5 and 6a). Of the eleven α-helices and 18 β-sheets, six α-helices and twelve β-sheets comprise the large domain. Among them, six β-sheets (β1, β2, and β15–β18) and six β-sheets (β5, β6, and β11–β14) face each other at the center of the large domain, to form a mixed parallel and antiparallel β-strand structure, surrounded by α-helices (α1–3, α7, α8, and α11) to create the αββα barrel architecture (Fig. 3a). Of the remaining α-helices and β-sheets, two α-helices (α9 and α10) participate in the construction of the C-domain. This domain contains several long loops and partially covers one side of the top of the αββα architecture in the large domain. The G-domain is composed of two α-helices (α5 and α6) and four antiparallel β-sheets (β7–β10), and is attached to almost the opposite site of the C-domain on the top of the αββα architecture. Two α-helices (α5 and α6) of the G-domain on the monomer protrude toward another dimer-forming monomer, and participate in the dimer formation together with part of the outer surface of the αββα core structure (Fig. 3c). In addition, the G-domain is encapsulated by the C-domain of the other monomer via several contacts, such as α9' of the

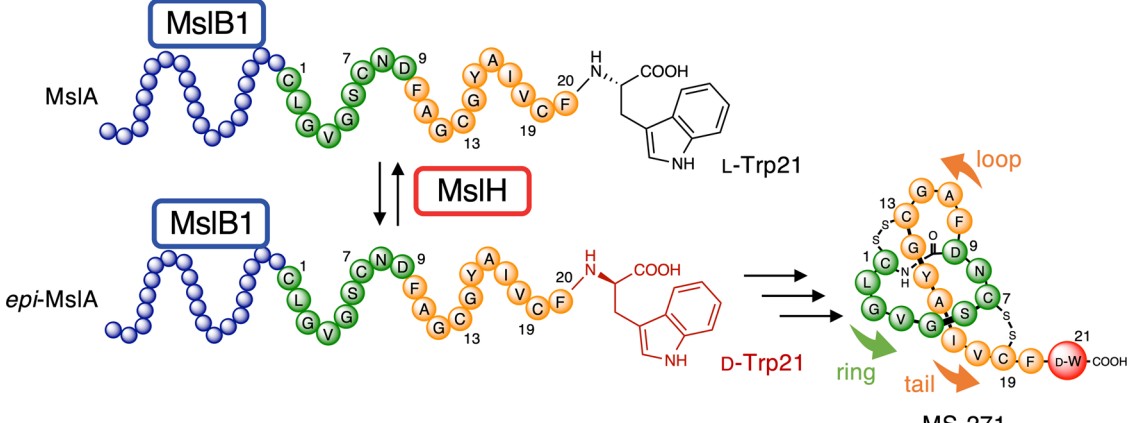

**Fig. 1 | Epimerization reaction catalyzed by MslH in the biosynthesis of MS-271.** The leader peptides of the MslH substrates, MslA and *epi*-MslA, are shown in blue, while their core peptides are highlighted in green and orange in accordance with

the formation of the ring and loop-tail architectures in MS-271, respectively. MslB1 binds to the leader peptide region.

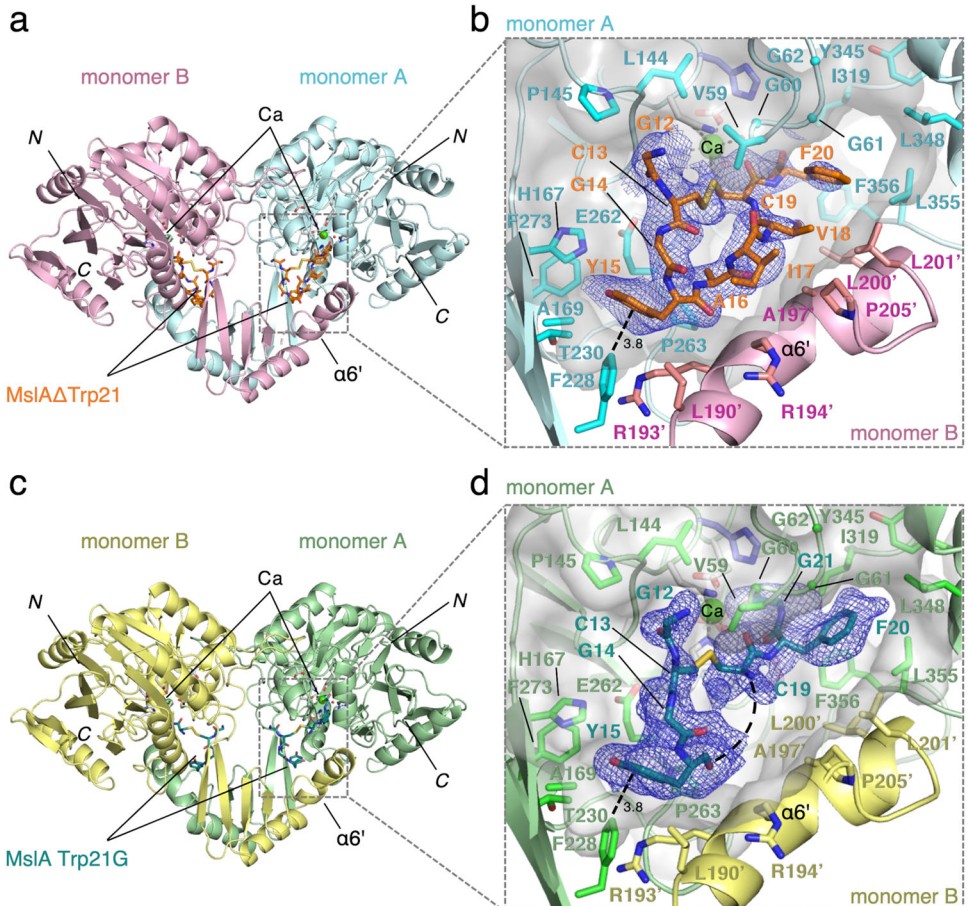

**Fig. 2 | Crystal structures of MslH:MslAΔTrp21 and MslH:MslA Trp21G.** Overview of the dimeric MslH:MslAΔTrp21 structure (**a**) and the MslH:MslA Trp21G structure (**c**). Close-up view of the MslH-MslA analogue binding cleft with the surface model in the MslH:MslAΔTrp21 structure (**b**) and that of the MslH:MslA Trp21G structure (**d**). A representative OMIT electron density map (mFo-DFc) contoured to 2σ around the MslA analogue is shown by the blue-colored mesh. The straight dashed lines represent the distances in Å. The curved dashed line represents the missing MslA Ala16-Val18 residues.

C-domain in the other monomer with the long loop structure between β6 and β7 of the G-domain (G-loop) (Fig. 3a, c). As a result, α6′ from another monomer, together with the four antiparallel β-sheets on the G-domain and part of the C-domain of the monomer, forms a large cleft on the edge of the αββα architecture in the core structure (Fig. 2).

Structure-based similarity searches revealed that MslH possesses a calcineurin-like fold[25] as part of its structure, as previously predicted by a Phyre2 analysis[19]. The overall structure of the MslH:apo structure is most similar to that of the metal-dependent phosphodiesterase YmdB in *Bacillus subtilis* (PDB ID: 4B2O)[26], with an RMSD value of 2.26 Å for 71 Cα atoms (corresponding to 16% of the amino acids of MslH), even though their amino acid sequence identity is only 16%. The structural similarity is mainly observed in the αββα architecture (Fig. 3b and Supplementary Figs. 6 and 7). A structural comparison of both proteins revealed that the two β-strands (β17 and β18) and one α-helix (α11) structures on the *C*-terminal side were characteristic of MslH, among the αββα architectures. Notable differences between the MslH and YmdB structures were observed in the C-domain and G-domain in MslH (Fig. 3a, b and Supplementary Fig. 7). In these structures, the C-domain in MslH is a steric homolog of the structure consisting of loops and α6 in YmdB, although that of MslH comprises loops with two α-helices, α9 and α10. In contrast, the G-domain in MslH is absent in YmdB.

## Active-site architecture of MslH
The electron density of MslAΔTrp21 in the MslH:MslAΔTrp21 crystal structure allowed us to locate a cyclic peptide molecule (cMslA Gly12-

Phe20), consisting of the Gly12-Phe20 part on the *C*-terminal side of MslAΔTrp21 with a disulfide bond between Cys13 and Cys19, in the large cleft conformed by two monomers (Fig. 2a, b). However, no electron density corresponding to either the MslA leader peptide or MslB1 was identified in the MslH:MslAΔTrp21 crystal structure. The cMslA Gly12-Phe20 of MslAΔTrp21 molecule is accommodated in the cleft with various interactions, such as a σ–π interaction between Phe228 of monomer A and Tyr15 in cMslA Gly12-Phe20 (3.8 Å), hydrophobic interactions between α6′ of monomer B and MslA Ala16-Val18, and polar interactions between Gly60 (3.4 Å)/His295 (2.7 Å) in MslH and the carboxyl group of the *C*-terminal MslA Phe20 of MslAΔTrp21 (Figs. 2b and 4a and Supplementary Fig. 8a). In a manner similar to the MslH:MslAΔTrp21 structure, the MslH:MslA Trp21G structure accommodated the MslA Trp21G molecule, consisting of the Gly12-Tyr15 and Cys19-Gly21 parts on the *C*-terminal side of MslA Trp21G with a disulfide bond between Cys13 and Cys19, in the large cleft (Fig. 2c, d and Supplementary Figs. 8b and 9a, b). In addition, the amide nitrogen and oxygen atoms between Phe20 and Gly21 in MslA Trp21G formed polar interactions with the oxygen atom in the backbone of MslH Gly60 and the nitrogen atom of the His295 side chain, respectively (2.6 Å and 2.5 Å) (Fig. 4b). Because the amide bond was recognized by MslH in a similar manner to the carboxyl group of the *C*-terminal MslA Phe20 (Supplementary Fig. 9c), the electron densities of MslA analogues likely represented the binding mode of the original MslA substrate in MslH. However, the disulfide bond observed between Cys13 and Cys19 in the MslA analogues is not present in the biosynthesized final product, MS-271, which has disulfide linkages

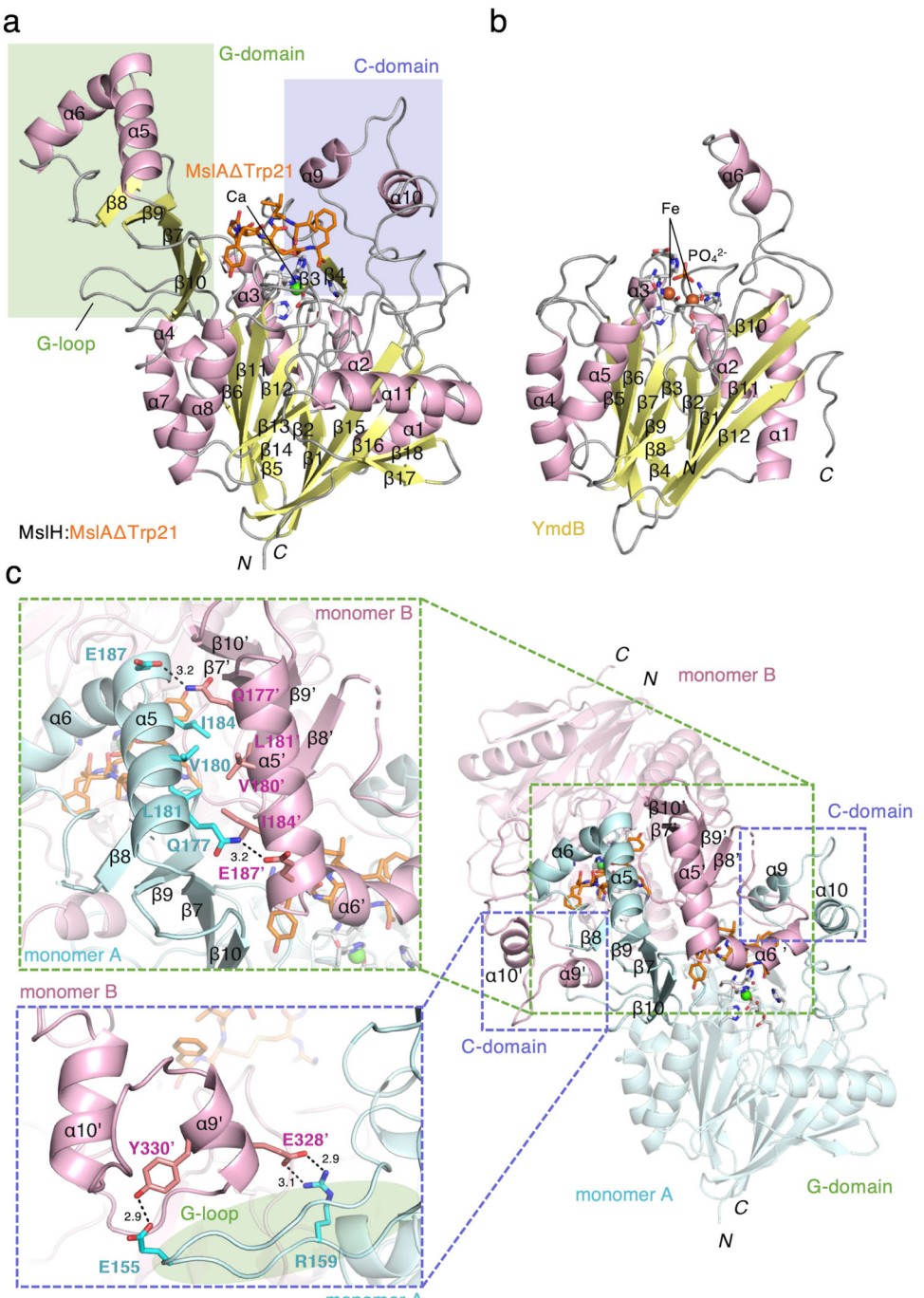

**Fig. 3 | Overview of the MslH:MslAΔTrp21 structure.** Stereographic ribbon illustrations of the MslH:MslAΔTrp21 monomer structure (**a**) and the reported YmdB monomer structure (PDB ID: 4B2O) (**b**)[26]. α-Helices and β-sheets are colored pink and yellow, respectively. The secondary structural elements are labeled according to each topology. The residues and metal ions in the active-site environment are shown as sticks colored by atom type and spheres, respectively. The carbon-backbone of MslAΔTrp21 is colored orange. **c** Overview of dimer formation focused on the C-domain and G-domain.

between Cys1-Cys13 and Cys7-Cys19 (Fig. 1). It is hypothesized that during MS-271 maturation, the core peptide of MslA generates disulfide linkages between Cys1-Cys13 and Cys7-Cys19. To determine indirectly whether the MslA analogues observed in the MslH:MslAΔTrp21 and MslH:MslA Trp21G crystal structures would be bound to the original substrate-binding site, we performed in vitro assays with and without DTT. As a result, the addition of DTT to the MslH reaction had no effect on the MslH catalytic activity (Supplementary Fig. 10). Furthermore, even MslA variants with serine residues at Cys13 and Cys19 (MslA Cys13/19S), respectively, were competent substrates in the MslH reaction, and exhibited the same activity as in

the case of the intact MslA substrate (Supplementary Fig. 10). These observations suggest that the disulfide linkage between Cys13 and Cys19 in MslA analogues in the MslH structures is not necessary for the catalytic activity of MslH. Thus, the original substrate would also be accepted in the same cleft as the MslA analogues observed in the MslH crystal structures, and the catalytic reaction site might occur around Gly21 on the MslA Trp21G molecule.

The structure analyses also revealed that the *C*-terminal MslA Phe20 carboxyl group in MslAΔTrp21 and the *C*-terminal MslA Phe20 and Gly21 in MslA Trp21G are located near a hydrophobic pocket consisting of MslH Gly60-62 and Leu64 on the core domain and Ile319,

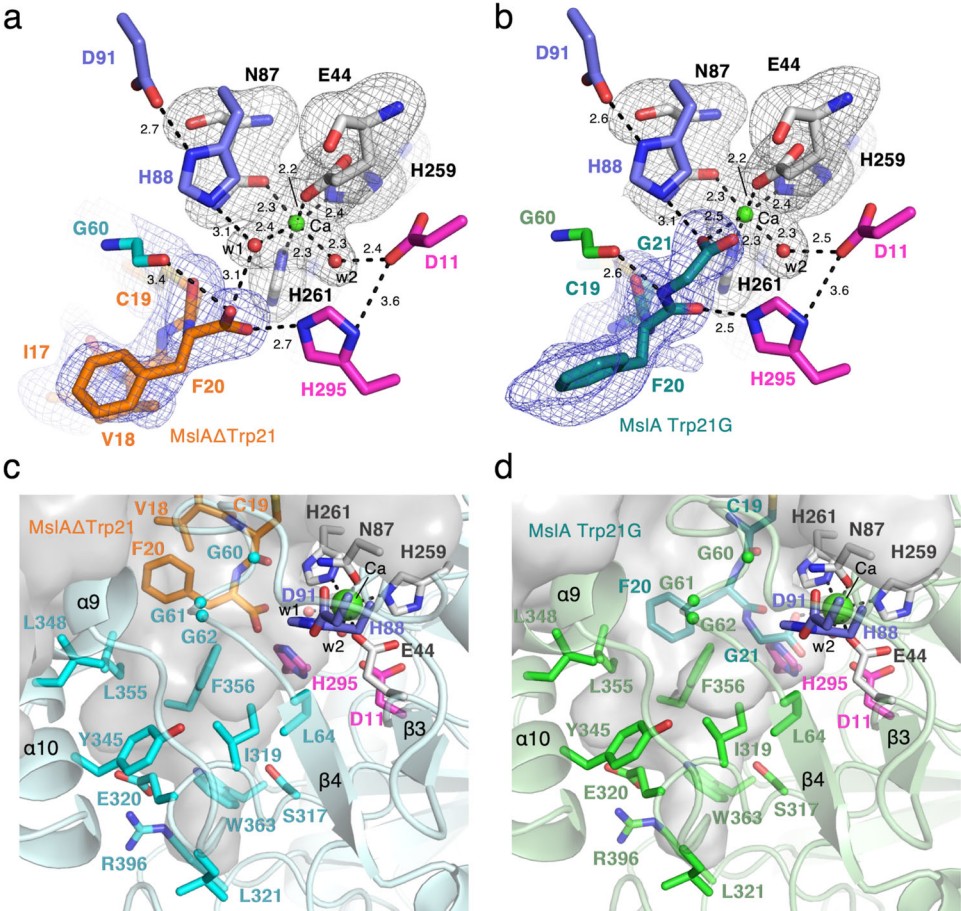

**Fig. 4 | Close-up view of the catalytic site.** Close-up view of the metal-coordination site in the MslH:MslAΔTrp21 structure (**a**) and MslH:MslA Trp21G structure (**b**). A representative OMIT electron density map (mF$_o$-DF$_c$) contoured to 2σ around MslA analogues is shown by the blue-colored mesh. Another representative OMIT electron density map (mF$_o$-DF$_c$) contoured to 3σ around water molecules, metal-coordinated residues, and Ca(II) is shown by the gray-colored mesh. The dashed lines represent the distances in Å. Surface representation of the hydrophobic pocket around the *C*-terminus of MslA analogues in the MslH:MslAΔTrp21 structure (**c**) and MslH:MslA Trp21G structure (**d**).

Tyr345, Leu348, Leu355, and Phe356 on the C-domain in monomer A at the edge of the cleft, respectively (Fig. 4c, d and Supplementary Figs. 8 and 9d). In addition, an architecture that would construct a six-coordination motif, consisting of four amino acid residues Glu44, Asn87, His259, and His261 and two water molecules (w1 and w2), was detected around the *C*-terminal carboxyl group of MslA Phe20 in the MslH:MslAΔTrp21 crystal structure (Fig. 4a and Supplementary Fig. 11a, b). This six-coordination motif is conserved in the MslH:apo and MslH:MslA Trp21G structures (Supplementary Figs. 12 and 13), although one of the oxygen atoms of the *C*-terminal carboxyl group of MslA Gly21 occupied the space used by the w1 water molecule in the MslH:MslA Trp21G structure (Supplementary Figs. 9c and 13c), and is located at the entrance of the central ββ barrel behind the cleft. In the six-coordination, the w2 water molecule hydrogen-bonds with the side-chain of Asp11 and is further held in the structure by the steric contraction of the His295 side-chain, which forms a hydrogen-bond with Asp11 (Fig. 4a, b). The other water (w1) in the MslH:MslAΔTrp21-structure and one of the oxygen atoms of the *C*-terminal carboxyl group of MslA Gly21 in the MslH:MslA Trp21G structure are located near His88 forming a hydrogen-bond with Asp91, respectively (Fig. 4a, b). We also found some remaining electron density, which was obviously different from those of the peptides and water molecules, in the center of the six-coordination system (Supplementary Fig. 14). When another water molecule (w3) was added to the observed electron density at the center of the six-coordination site, the Fo-Fc map remained positive around the added water molecule, indicating the

presence of an unidentified metal ion instead of a water molecule (Supplementary Fig. 14a), at a distance of 2.5 Å from the oxygen atom of the Gly21 carboxyl group (Fig. 4b). These observations suggested that the catalytic center responsible for the epimerization reaction of MslH is in this region.

In order to address this possibility, we first performed docking simulations with cMslA Gly12-Trp21, a Gly12-Trp21 peptide of MslA with a disulfide linkage between Cys13 and Cys19, to determine if this hydrophobic pocket is large enough to accommodate the indole side-chain of the *C*-terminal residue Trp21 of wild-type MslA. The initial model of cMslA Gly12-Trp21 was designed, based on the cMslA Gly12-Phe20 architecture in the MslH:MslAΔTrp21 structure. We selected several amino acid residues contributing to the hydrophobic pocket flexibility, and ran AutoDock Vina docking simulations using the designed cMslA Gly12-Trp21 peptide as the initial phase. The docking study provided a cMslA Gly12-Trp21 model with peptides bound in the cleft formed by two monomers, in almost the same location and orientation as those of MslAΔTrp21 and MslA Trp21G in the MslH:MslAΔTrp21 and MslH:MslA Trp21G structures, respectively. As expected, the indole ring of Trp21 was accommodated in the hydrophobic pocket (Fig. 5 and Supplementary Fig. 15). In the model structure, the side-chain of Phe20 in the cMslA Gly12-Trp21 molecule formed a parallel-displaced π-stacking interaction (2.7 Å) with the indole ring of its own *C*-terminal Trp21. Especially, the backbone structure containing the carboxyl group of MslA Trp21 is predicted to bind at a location consistent with that of MslA Gly21 in the MslH:MslA

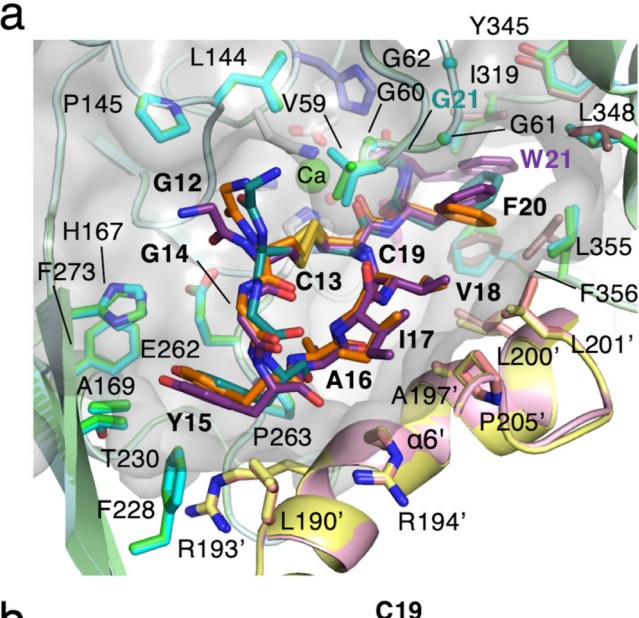

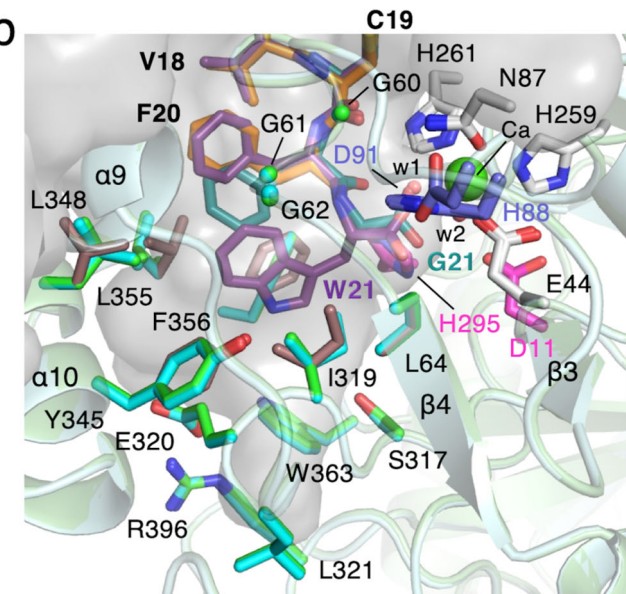

**Fig. 5 | Docking simulation of cMslA Gly12-Trp21.** Surface representation of the binding cleft for MslA (**a**) and the proposed MslA Trp21 binding pocket (**b**) in the simulated MslH:cMslA Gly12-Trp21 structure (carbon-backbone of modeled MslH: light purple; carbon-backbone of modeled MslA Gly12-Trp21: dark purple) and its superimposed views with the MslH:MslAΔTrp21 crystal structure (carbon-backbone of MslH: cyan and pink; carbon-backbone of cMslA Gly12-Phe20: orange) and the MslH:MslA Trp21G crystal structure (carbon-backbone of MslH: green and yellow; carbon-backbone of cMslA Gly12-Gly21: blue green).

Trp21G complex structure, and the carboxyl oxygen atom of the MslA Trp21 is located at a 3.0 Å distance from the unidentified metal ion (Fig. 5b and Supplementary Fig. 16).

Next, we investigated the metal-binding site in MslH by comparing the structures of MslH and the structurally closest metal-dependent phosphodiesterase, YmdB[26]. This comparison indicated that the residues and water molecules involved in the formations of the six-coordination motif and related hydrogen-bond networks from the water molecules in the MslH:MslAΔTrp21 crystal structure are located at positions very similar to those in the Fe(II) metal-binding-site consisting of Asp8, Glu39, Asn40, Asn67, His150, His175, His177, His68, Asp71, and a water molecule of YmdB (Supplementary Fig. 11). In particular, the positions of Asn87 and His259 lining the six-coordination motif and His88 and Asp91 forming the hydrogen-bond networks from

the water molecules in MslH are nearly identical to those of Asn67 and His150, and His68 and Asp71 of YmdB, respectively (Supplementary Fig. 11d). Although the side-chain of Asp11 in MslH was slightly rotated as compared with that of Asp8 in YmdB, both residues are located in almost the same position. The main-chain of His261 in MslH, corresponding to His175 in YmdB, has a significantly different position from that of His175 in YmdB, due to the substantial conformational change between the loop containing His261 in MslH and the corresponding loop containing His175 in YmdB. Meanwhile, the side-chain of His261 is also in almost the same position as that of His175 in YmdB, to participate in the six-coordination. In contrast, MslH lacked sufficient space to bind the second Fe(II) found in YmdB, presumably due to several conformational changes such as the lack of the amino acid residue at the location occupied by Asn40 in YmdB, the rotational and positional changes of the side-chain of Glu44 corresponding to Glu39 in YmdB, and the rotation of the side-chain of His295 in MslH by 90 degrees, as compared with that of His177 in YmdB. Thus, the MslH crystal structure suggested that MslH possesses a rearranged structural analogue of the Fe(II)-binding site in YmdB. These rearrangements were also conserved in the MslH:apo and MslH:MslA Trp21G structures (Supplementary Figs. 12 and 13).

**Ca(II)-binding of MslH**

To further confirm the importance of the observed six-coordinated motif during the enzyme reaction, we performed mutagenesis studies of MslH, in which four residues, Glu44, Asn87, His259, and His261 in the six-coordination motif were each substituted with alanine. The mutagenesis studies revealed that the MslH Glu44A and Asn87A variants lacked the epimerase activity and the MslH His259A and His261A mutants diminished the activity, suggesting the crucial roles of the six-coordination motif for its catalysis (Fig. 6).

Meanwhile, we performed ICP-MS analyses of purified MslH to clarify the metal-dependency in the MslH catalytic mechanism. The ICP-MS measurements unveiled the presence of Ca(II) and Mn(II) ions at ratios of 0.4 mole and 0.3 mole per mole of MslH, respectively, suggesting either the Ca(II)- or Mn(II)-dependence of MslH (Supplementary Table 3). Presumably, these metal ions were incorporated into MslH during enzyme expression in *E. coli*. Thus, to identify the metal ion responsible for the MslH epimerase activity, we expressed MslH in *E. coli* cultured in M9 medium containing only NaCl, MgSO₄, and CaCl₂, which are essential metals for *E. coli* growth. The in vitro enzymatic reaction indicated that the catalytic activity was maintained in MslH expressed in M9 medium, as in the case of the recombinant MslH expressed in Luria-Bertani (LB) medium (Supplementary Fig. 17). In addition, to gain information about the metal ions integrated into the crystal structure, the MslH:MslAΔTrp21 crystal structure refinements were performed by accommodating Ca(II), Mg(II), Mn(II), and Fe(II) ions in the aforementioned remaining electron density in the six-coordination motif. The Fo-Fc density maps displayed a positive value for Mg(II) and negative values for Mn(II) and Fe(II), but were neither positive nor negative for Ca(II) (Supplementary Fig. 14b–e). Since the *B*-factor values of metal-coordinating amino acid residues and water molecules are typically near the metal's *B*-factor value, we also investigated their *B*-factor values. Consequently, the *B*-factor value of the Ca(II) ion in the crystal structure was the closest to the average of those of the metal-coordinating residues and water molecules, while that for Mg(II) was below and those for Mn(II) and Fe(II) were above the average (Supplementary Table 4). Taken together, the aforementioned data allowed us to conclude that MslH is a Ca(II)-dependent epimerase.

**Catalytic mechanism of MslH**

As previously reported, MslH catalyzed the reversible epimerization of the L- and D-amino acid orientations of MslA Trp21 in in vitro reactions[19]. To further support the reversible epimerization of MslH, we also performed the in vitro MslH reaction using MslA in D₂O, to

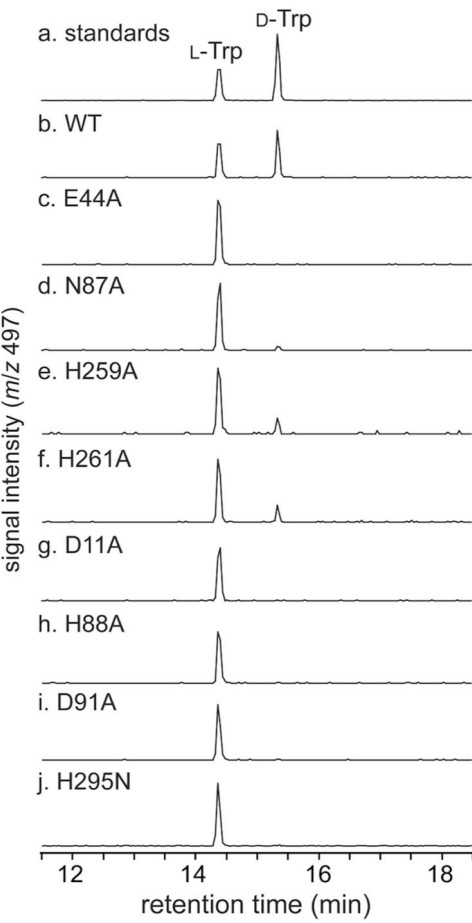

**Fig. 6 | LC-MS analysis (ESI-negative ion mode, *y*-axis represents signal strength measured at *m/z* 497, and the same scale applies to all chromatograms) of L-FDLA–Trp prepared for the in vitro reaction of MslH. a** L-Trp and D-Trp standards, **b** wild-type MslH (WT), **c** MslH Glu44A variant, **d** MslH Asn87A variant, **e** MslH His259A variant, **f** MslH His261A variant, **g** MslH Asp11A variant, **h** MslH His88A variant, **i** MslH Asp91A variant, and **j** MslH His295N variant.

monitor proton abstraction and addition at the Cα of the Trp21 residue. The subsequent LC-MS analysis revealed that MslH produced not only MslA with *C*-terminal deuterated D-Trp, but also deuterated L-Trp (Supplementary Fig. 18). Thus, MslH catalyzes the reversible epimerization of the *C*-terminal L-Trp21 residue of MslA in a Ca(II)-dependent manner. Therefore, a configuration facilitating the epimerization might be located near the L-Trp21 of the substrate in MslH, separately from the six-coordination. The crystal structures of MslH:MslA Trp21G and the docking model of MslH:MslA Trp21 suggested that His88 coupled with Asp91, corresponding to the YmdB catalytic residues, His68 and Asp71[26], was near the α-proton of the L-Trp21 model, while His295 coupled with Asp11, corresponding to the YmdB second Fe(II) ion-coordinating His177 and Asp8, was opposite from the α-proton, and these histidine residues were positioned across the Cα centers of the MslA Gly21 and MslA Trp21 residues in a linear manner, respectively (Fig. 7). Similar cases were also found in other epimerases that catalyze reversible epimerization. For examples, the catalytic process of the PLP-independent epimerase, *ap*MurI[27], and a Mg(II)-dependent epimerase, AE epimerase[28], enable reversible D- and L-epimerization of an amino acid by positioning the two catalytic cysteine residues and the two catalytic lysine residues against the reaction point of the substrate in a linear manner, respectively. Therefore, the observed linear configuration of the histidine residues coupled with aspartate residues would allow MslH to facilitate the reversible epimerization of the L- and D-amino acid forms of MslA Trp21.

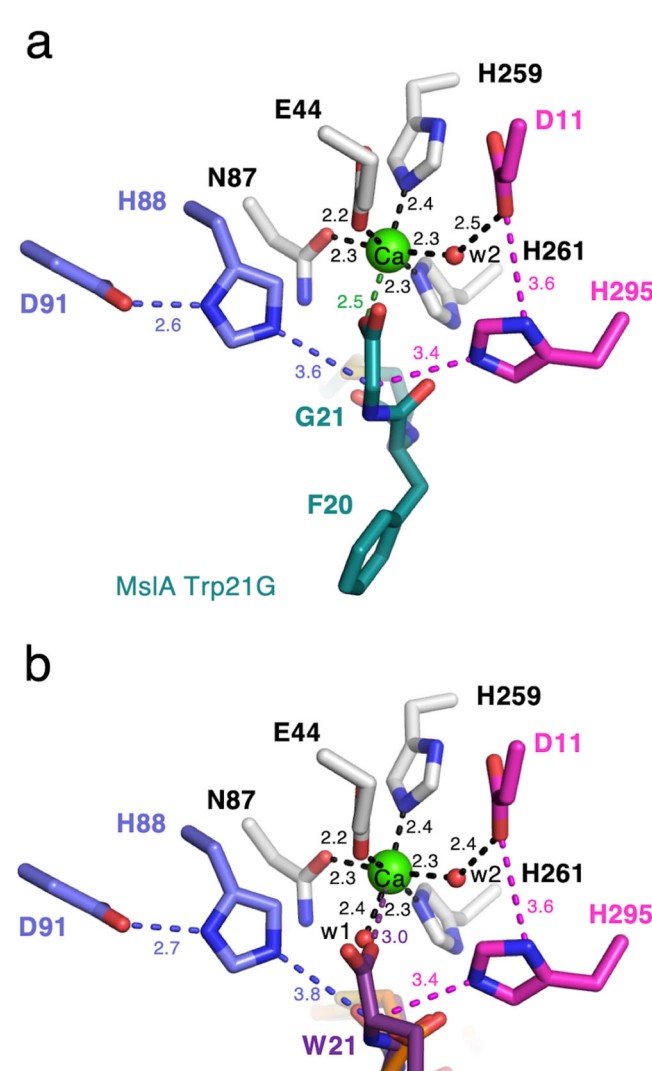

**Fig. 7 | Linear configuration of the histidine residues around the Cα center of the *C*-terminal MslA residue.** The blue and magenta dashed lines represent the distance between the Nε histidine atoms and the Cα center of MslA Gly21 in the MslH:MslA Trp21G complex structure (**a**) and the distance of the simulated MslA Trp21 in the cMslA Gly12-Trp21 docking model superimposed with the MslH:MslA ΔTrp21 crystal structure (**b**) with the polar interaction between the Nδ histidine atoms and one of the Oδ aspartate atoms (distances in Å). Other focused distances are also represented.

The two pairs of His88/Asp91 and His295/Asp11 residues observed in MslH presumably act as the catalytic residues. Hence, we conducted additional mutagenesis studies on the four residues (MslH Asp11, His88, Asp91, and His295). In the mutagenesis studies, all residues except for His295 were substituted with alanine, while His295 was substituted with asparagine, with the same size as histidine, because of the instability of the His295A variant during the purification and enzyme reaction. The in vitro experiment revealed that all variants lacked epimerase activity, indicating that the four residues are crucial for catalysis (Fig. 6). Considering that His88/Asp91 in MslH correspond to the YmdB catalytic residues, His68 and Asp71[26], the loss of the MslH activity by the His88 and Asp91 point mutations explains the functions of His88/Asp91 as catalytic residues. In contrast, the loss of catalytic

**Fig. 8 | Schematic representations of the proposed catalytic mechanism of MslH.** MslH utilizes two pairs of His/Asp catalytic residues to facilitate the reversible epimerization of the *C*-terminal Trp21 of MslA.

activity of the His295N and Asp11A variants could be due to a structural collapse of the metal-binding and substrate-binding sites, caused by a breakdown of the hydrogen bonding network between six-coordination motif and Asp11/His295 via the w2 water molecule and substrate (Fig. 4a, b). Therefore, we evaluated the His295N and Asp11A variants from a structural aspect. We obtained their crystal structures, MslH Asp11A:apo and MslH His295N:apo at 2.41 and 2.55 Å resolutions, respectively (Supplementary Table 2). Both crystal structures indicated that the variants retained the same six-coordination motif as that in the wild-type MslH crystal structure (Supplementary Figs. 19 and 20). The His295N substitution had no effect on the active center organization, while the Asp11A substitution led to the flexibility of His295 in the MslH Asp11A:apo crystal structure, where a 1:1 alternative state was observed for the His295 residue (Supplementary Fig. 19). The main chain carbonyl group of the Pro294 residue in the MslH Asp11A:apo crystal structure was inverted by 180 degrees, as compared with that of the wild type (Supplementary Fig. 19e). Moreover, the comparison of the MslH His295N:apo structure with the MslH:MslA Trp21G complex structure showed that the distance between the Cα center of MslA Gly21 and Oδ of Asn295 atom is 4.5 Å, which is longer than that (3.4 Å) in the MslH:MslA Trp21G structure (Fig. 7a and Supplementary Fig. 20f). These observations suggested that the Asp11 and His295 residues would play important roles in maintaining the appropriate catalytic state of MslH.

Based on the comprehensive data described above and previous reports, we propose that MslH employs an "acid/base" chemistry similar to that of UDP-galactose 4-epimerase[29], in the epimerization reaction to generate *epi*-MslA, as follows (Fig. 8): MslH accommodates the Gly12-Trp21 portion of the MslA/MslB1 complex substrate within the cleft, and initiates the α-proton abstraction of L-Trp21 in the MslA/MslB1 complex by Asp91-activated His88, to produce an enolate intermediate in a transition state with the Ca(II) ion. Subsequently, a proton transfer from the His295 side-chain, located on the opposite side, occurs on the α-position of the enolate intermediate to produce *epi*-MslA via the tautomerization of the enolate form to the keto form of the intermediate, and the His295 side-chain further abstracts a proton from the Asp11 side-chain. Finally, MslH releases *epi*-MslA as the end product, which is subsequently subjected to the lasso cyclization steps, and recovers the initial state of His88-Asp91 by proton transfer starting from His295 to Asp91, via the Asp11-w2-Ca(II)-w1-His88 network observed in the crystal structure. In contrast, when MslH accepts *epi*-MslA as the substrate, the enzyme facilitates the epimerization reaction of *epi*-MslA to MslA, by employing Asp11-activated His295 as the base catalyst (Fig. 8). As a side note, although the initial proton abstraction may occur with His295, His88 most likely acts as the initial catalytic base, based on our docking study (Fig. 7b).

## Discussion

The MslH epimerase is involved in the first stage of modification in the biosynthesis of the lasso peptide MS-271, making its functional investigation a necessity in the expansive field of lasso peptide research (Fig. 1). A prior functional analysis did not completely elucidate its mechanism, and MslH was described as an unusual enzyme that does

not require any cofactors[19]. Herein, the X-ray crystal structure analysis of MslH demonstrated that it possesses a calcineurin-like fold[25] and catalyzes the epimerization reaction in a Ca(II)-dependent manner. To the best of our knowledge, this type of epimerase has never been found in the realm of RiPPs, and this report demonstrates the relationship to calcineurin-like fold and epimerization.

Our crystallographic analysis of MslH in association with its substrate MslA analogues revealed that it accomplishes the recognition of the substrate precursor peptide and the epimerization of the *C*-terminal amino acid by employing several characteristic structural architectures, which have not been found in other enzymes with calcineurin-like folds. The previously reported enzymes with calcineurin-like folds, including calcineurin, contain two metals in their active centers for their catalytic activities[24,25]. In contrast, MslH adopted only one metal ion in the active site, even though it forms a conserved calcineurin-like fold. The MslH crystal structure revealed that this is due to the conformational rearrangement of the residues involved in and surrounding the metal-binding region in MslH, especially Asp11, Glu44, His261, and His295, as compared with those in the metallo-dependent phosphatases with calcineurin-like folds, such as YmdB (Supplementary Fig. 11d)[26]. Subsequently, the key amino acids Glu44, Asn87, His259, and His261 are directly coordinated to the Ca(II) ion, and the two sets of catalytic dyads, His88/Asp91 and His295/Asp11, were defined by the site-directed mutagenesis and docking studies. These residues are well conserved in the other MslH-like enzymes, including *Streptomyces* sp. M-271, *S. nodosus*, and *S. olivochromogenes* (Supplementary Figs. 21 and 22), suggesting that the MslH enzymes catalyzing Ca(II)-dependent epimerization reactions would be universally employed in *Streptomyces*.

Our present work also indicated that MslH could facilitate the first α-proton abstraction of MslA L-Trp21 by using His88, corresponding to the widely conserved histidine residue in the active centers of metallophosphoesterases[25,30]. Moreover, MslH would employ the His88/Asp91 dyad, corresponding to the His68/Asp71 catalytic dyad in YmdB, as the acid/base catalytic residues[26]. However, His150/Asn67 in YmdB, corresponding to another possible His295/Asp11 catalytic dyad responsible for the proton transfer in MslH, was reportedly utilized for the second metal coordination in metallophosphoesterases[25]. This difference was caused by the loss of the second metal coordination in MslH, and thereby, only the His295/Asp11 dyad in MslH presumably conferred the ability to catalyze the proton transfer for the epimerization (Fig. 7). Thus, the crystal structure analysis of MslH has provided insight into the diversity of the metal-binding sites in enzymes with calcineurin-like folds.

The structural architecture in the MslH active site, combined with labeling tests using heavy water, suggested that the two His/Asp pairings promote a reversible acid-base reaction. This reverse reaction is anticipated to be initiated by the α-proton abstraction of D-Trp21 by the His295/Asp11 dyad. Interestingly, the lasso peptide MS-271 with a *C*-terminal L-Trp has never been isolated from a natural source. Furthermore, the previous MS-271 expression study in *Streptococcus lividans* TK23 with the *msl* gene cluster demonstrated that MS-271 and its derivatives were not produced in the absence of the *mslH* gene[22]. In

light of these facts, the biosynthetic enzyme(s) following the MslH reaction would strictly regulate the *C*-terminal D-Trp form.

Based on several approaches, we demonstrated that MslH would be matured with a bound Ca(II) ion. To the best of our knowledge, the heavy divalent metals such as Fe(II), Zn(II), and Mn(II) have been identified in the metal-binding sites of enzymes with calcineurin-like folds[25], but except for MslH, no other enzyme with a calcineurin-like fold employing Ca(II) as the cofactor has been reported. In our previous experiments, we could not identify the metal-dependency of MslH because the addition of ethylenediaminetetraacetic acid (EDTA) had no effects on its activity[19]. However, the MslH crystal structure indicated that the metal-binding sites are buried deep inside the protein (Supplementary Fig. 23), thus blocking EDTA access to the metal ions.

Another noteworthy point is the multimeric structure of MslH for the precursor peptide recognition, which is not found in other enzymes with calcineurin-like folds. The crystal structure analysis of MslH revealed that the subdomains (C- and G-domains) were characteristically involved in interactions with other monomers to form dimers as well as the substrate-binding cleft on each dimer, and the dimers formed an octamer composed of eight monomers (Fig. 2 and Supplementary Fig. 4). Since other monomers do not obstruct these clefts, each monomer might simultaneously react with the MslA-MslB1 complex. However, the leader peptides of MslA and/or the complexed protein MslB1 were unclear, suggesting a weak interaction between the MslA-MslB1 complex site and MslH. In actuality, the docking simulations performed on cMslA Gly12-Trp21 revealed a striking resemblance to cMslA Gly12-Phe20 in the MslH crystal structure as the most stable form (Fig. 5 and Supplementary Fig. 15), indicating that the core peptide on the *C*-terminal side of MslA binds to the enzyme in a stable manner. These observations suggest that the substrate recognition by MslH is mainly performed in the MslA core peptide region, and instead of primarily facilitating the substrate recognition of MslH, the MslA leader peptide/MslB1 complex structure supports the substrate recognition of MslH by inducing a relaxed state of the MslA core peptide. As discussed above, our X-ray crystallographic analysis has provided some clarification of the binding mode of the *C*-terminal MslA core peptide in the MslH structure. Our observations will facilitate the future production of non-natural lasso peptides with *C*-terminal D-amino acids, by the enzymatic engineering of the amino acid residues involved in the substrate recognition by MslH.

## Methods

### General information

Unless otherwise noted, all reagents were from commercial sources, including Fujifilm Wako Pure Chemical Corporation, Ltd. (Osaka, Japan), Sigma-Aldrich Japan (Tokyo, Japan), and Hampton Research Co. (Laguna Niguel, CA) (Supplementary Table 6). Enzymes and kits for DNA manipulations were purchased from Takara Bio (Shiga, Japan), Nippon Gene Co. Ltd. (Tokyo, Japan), New England Biolabs Japan Inc. (Tokyo, Japan), and Toyobo Co. Ltd. (Osaka, Japan). Polymerase chain reaction (PCR) was performed with a GeneAmp PCR System 9700 thermal cycler and Tks Glex DNA polymerase (Takara Bio). DNA sequencing was performed by Fasmac (Kanagawa, Japan). *E. coli* was subjected to general genetic manipulations in accordance with standard protocols[31]. Plasmids in *E. coli* were maintained using appropriate antibiotics at the following concentrations: ampicillin (100 μg/ml), kanamycin (25 μg/ml), chloramphenicol (30 μg/ml), and/or streptomycin (20 μg/ml). Milli-Q® Ultrapure (MQ-grade) water was used for buffers. The LC-MS analysis was performed with a Waters ACQUITY UPLC system equipped with an SQ Detector 2, using a Mightysil RP-18GP Aqua column (150 × 2.0 mm, 3 μm). Protein purifications were performed using an ÄKTA Explorer FPLC and ÄKTA Purifier FPLC system (Cytiva, Marlborough, MA).

### Protein production and purification

The *N*-terminally His$_6$-tagged MslH and MslA-MslB1 complexes (wild-type and all variants) were expressed in *E. coli* BL21 (DE3) cells, using

the pET28a-His-MslH and pET21-SKIK-His-MslA-pCDF-MslB1 plasmids, respectively[19]. All protein expression was performed in the LB medium. The composition of the M9 medium for expression was 22.1 mM KH$_2$PO$_4$, 8.5 mM NaCl, 18.7 mM NH$_4$Cl, 47.8 mM Na$_2$HPO$_4$, 2.0 mM MgSO$_4$, 22.4 mM D-glucose, 0.1% (v/v) glycerol, 0.2 mM CaCl$_2$, 0.02% (w/v) thiamin, 0.02% (w/v) thymidine, 0.02% (w/v) adenosine, 0.02% (w/v) guanosine, and 0.02% (w/v) cytidine. The cells were harvested, resuspended in buffer A (50 mM Tris-HCl, pH 8.0, 5% (v/v) glycerol, 100 mM NaCl, 25 mM imidazole), and then homogenized under high pressure to lyse the cells. The cell lysate was centrifuged (8000 × $g$, 30 min, 4 °C) to remove the insoluble cell debris. His$_6$-tagged proteins were purified using sequential Ni(II)-affinity chromatography (5 ml; Novagen; buffer: 30 CV of buffer A as wash buffer and 3 CV of buffer A containing 250 mM imidazole for elution), anion exchange column chromatography (RESOURCE Q 6 ml; Cytiva; buffer: 50 mM Tris-HCl, pH 8.0, 5% (v/v) glycerol, 0–1000 mM NaCl, 1 mM DTT), and size-exclusion chromatography (HiLoad 16/600 Superdex 200 pg column; Cytiva; buffer: 20 mM Tris-HCl, pH 7.5, 5% (v/v) glycerol, 5 mM NaCl, 1 mM DTT). Target protein-containing fractions were combined, and the SDS-PAGE analysis confirmed that MslH and MslA-MslB1 were both >95% pure (Supplementary Fig. 1). Purified proteins were kept at concentrations of 10 mg/ml (MslH) and 5 mg/ml (MslA-MslB1 complex) at −78 °C. Fresh aliquots were used for all MslH assays.

### In vitro peptide epimerization assay

A reaction mixture (150 μl) contained 470 μM MslA (SKIK- and His-tagged), 500 μM MslB1, and 5 μM MslH in buffer (100 mM Tris-HCl, pH 8.0, 300 mM NaCl, 10 mM DTT). Control reactions without DTT and reactions with mutants were also conducted. After incubation at 30 °C for 16 h, the reactions were terminated by adding 2×SDS-PAGE sample buffer (150 μl) and heating at 100 °C for 10 min. MslA in the reactions was purified with Tricine-SDS-PAGE (16%T, 3%C polyacrylamide gel), recovered from the gel with an Attoprep filter unit (ATTO), and rebuffered with 10 mM Tris-HCl (pH 8.0) using ultrafiltration (Amicon Ultracel-3, 0.5 ml, Merck). After hydrolysis of MslA at 110 °C for 10 h with 3 M 2-mercaptoethanesulfonic acid (MESA), the resulting hydrolysate was neutralized with 1 M aq. NaOH and tryptophan was purified by HPLC. The chiral derivatization using 1-fluoro-2,4-dinitrophenyl-5-L-leucinamide (L-FDLA) was performed with the following conditions: a reaction mixture contained 20 μl of 0.5 M sodium bicarbonate, 15 μl of sample solution, and 50 μl of 1% acetone solution of L-FDLA. After 2 h reaction at 37 °C, the reaction was quenched by addition of 10 μl of 1 M HCl followed by 300 μl of acetonitrile. Chiral analysis of the L-FDLA derivatives was performed on an ACQUITY UPLC system (Waters) with the following conditions: column, Mightysil RP-18GP Aqua column (150 × 2.0 mm, 3 μm); detection, ESI-negative mode on a SQ Detector2 and PDA; mobile phase, A: water containing 0.05% TFA, B: acetonitrile containing 0.05% TFA, 15% B for 0–2 min and a linear gradient to 95% B for 2–20 min; flow rate, 0.2 ml/min; column temperature, 40 °C. At least three independent reactions were performed for in vitro studies. To perform MslH reaction in D$_2$O, MslA, MslB1, and MslH were prepared in deuterated buffer (100 mM Tris-HCl, pH 8.0, 300 mM NaCl, 10 mM DTT in D$_2$O) using 0.5 ml Amicon Ultracel-3. A reaction mixture (150 μl) containing 470 μM MslA (SKIK- and His-tagged), 500 μM MslB1, and 5 μM MslH in deuterated buffer was first incubated at 30 °C for 1 h. After addition of MslH at final concentration of 10 μM, the reaction was further incubated at 30 °C for 15 h. The MslH reaction in D$_2$O was examined two times.

### Mutagenesis studies

Each full-length DNA encoding an MslH variant was amplified from the wild-type plasmid (pET28a-His-MslH) as the template by PCR, using Tks Gflex DNA Polymerase (Takara Bio) and the pairs of primers listed in Supplementary Table 5. Except for the Asp11A variant, overlap extension PCR was performed using primer pairs of *mslH*-F/variant specific-R primers and variant specific-F primer/*mslH*-R to amplify the *N*- and

*C*-terminal regions of *mslH*, respectively, for the first PCR, and the primer pair *mslH*-F/*mslH*-R was used for the second PCR. Each resulting DNA fragment was digested with *Nde*I and *Hind*III and cloned into the *Nde*I/*Hind*III sites of pET28a to produce *N*-terminally His$_6$-tagged fusion proteins. To construct the pET21-SKIK-His-MslA Cys13/19S variant, inverse PCR was performed using KOD-Plus-Neo DNA Polymerase (Toyobo), a pair of primers listed in Supplementary Table 5, and pET21-SKIK-His-MslA as the template. The PCR product was self-ligated with the In-Fusion cloning kit (Takara Bio), for expression as an *N*-terminally His$_6$-tagged fusion protein (pET21-SKIK for the MslA variant). To construct the pET21-SKIK-His-MslA ΔTrp21 and Trp21G, the primers pairs, *mslA*-F(*Sph*I)/*mslA*-ΔTrp21-R(*Hind*III) and *mslA*-F(*Sph*I)/*mslA*-Trp21G-R(*Hind*III), respectively, were used to amplify mutated PCR fragments from pET21-SKIK-His-MslA as the template. The resulting PCR fragments were cloned into the *Sph*I–*Hind*III sites of the pET21 vector. After confirmation of the sequences, pET28a-His-MslH variants were transformed into *E. coli* BL21 (DE3) and pET21-SKIK-His-MslA variants were transformed into *E. coli* BL21 (DE3) with pCDF-MslB1, according to the co-expression described above.

## ICP-MS analysis

For the metal ion content analysis, the buffer containing purified MslH was replaced with 100 mM Tris-HCl buffer (pH 8.0), and the protein was concentrated with an Amicon Ultra 3 K filter. The protein concentration was analyzed by an amino acid analysis, using the following procedure. The protein sample (50 μl) was hydrolyzed in 5.7 M hydrochloric acid containing 0.1% phenol (500 μl) at 100 °C for 24 h. After concentration in vacuo, the residue was dissolved in Milli-Q water (2000 μl) and an aliquot of the sample (40 μl) was analyzed with an L-8900 Amino Acid Analyzer (Hitachi High-Tech Co., Tokyo, Japan) at the Instrumental Analysis Division, Global Facility Center, Creative Research Institution, Hokkaido University. The concentration of MslH was determined based on the average amino acid concentration (Supplementary Table 3). For the ICP analysis, the protein sample (100 μl) was concentrated in vacuo and treated with 2 ml aq. nitric acid (69% (w/w), Ultrapur-100 grade, Kanto Chemical Co. Inc., Tokyo, Japan) at 100 °C. After 2 h, the sample was cooled to room temperature, mixed with 0.4 ml hydrochloric acid (35% (w/w), Ultrapur-100 grade, Kanto Chemical Co. Inc.), and diluted with Milli-Q water to 25.00 ml. This treated sample was analyzed with an Agilent 8800 Triple Quadrupole ICP-MS system (Agilent Technologies) in the collision (He and H$_2$) or mass-shift reaction mode (O$_2$) at the Open Facility Division, Global Facility Center, Creative Research Institution, Hokkaido University (Supplementary Table 3).

## Crystallography

The vapor diffusion method was used to develop crystals of *N*-terminally His$_6$-tagged MslH (5 mg/ml) in 500 nl sitting drops with 1:1 sample:precipitant ratios, at 20 °C (wild type) or 4 °C (Asp11A and His295N variants). The precipitants are listed in Supplementary Tables 1 and 2. For the co-crystallization with MslA Trp21G or MslAΔTrp21, the purified *N*-terminally His$_6$-tagged MslA Trp21G-MslB1 complex or the MslAΔTrp21-MslB1 complex was mixed with a solution of *N*-terminally His$_6$-tagged MslH (5 mg/ml) at a 1.2:1 MslA-MslB1:MslH mole ratio, 16 h prior to mixing with precipitants, respectively. Before cryo-cooling in liquid N$_2$, the crystals were cryo-protected using the mother liquor supplemented with 20% (v/v) glycerol. Data were collected at 100 K using synchrotron radiation at the Photon Factory (PF) beamline BL-1A. Data were indexed, integrated, and scaled using XDS[32] and Aimless in the CCP4 program package[33] (Supplementary Tables 1 and 2).

The MslH:apo crystal structure was solved by S-SAD, using the AutoSol[34] subroutine in PHENIX[35]. The MR method was used to determine the other MslH structures using the PHASER[36] subroutine in PHENIX[35] based on the MslH:apo crystal structure as the search model. The structural model was improved by iterative cycles of manual rebuilding in COOT[37] and crystallographic refinement in phenix.refine[38] (refinement details are summarized in Supplementary Tables 1 and 2). Crystal structure data have been deposited in the Protein Data Bank with PDB accession codes: 8GQ9 (MslH:apo), 8ITG (MslH:MslA Trp21G), 8GQA (MslH:MslAΔTrp21), 8GQB (MslH Asp11A:apo), and 8ITH (MslH His295N:apo). PyMOL[39] was used for the generation of graphical representations, and polder omit maps were calculated using Polder Maps[40] in PHENIX[35].

## Docking studies

The AutoDock Vina program[41] was used as the docking engine for the cMslA Gly12-Trp21 binding predictions. The starting position and simulation settings were obtained from the crystal structure of MslH:MslAΔTrp21. The initial structures of cMslA Gly12-Trp21 were based on the MslAΔTrp21 structure in the MslH:MslAΔTrp21 complex. The putative amino acid residues for binding of the indole ring of MslA Trp21, Leu64, Ile319, Tyr345, Leu348, Leu355, and Phe356, were chosen as the flexible residues. The calculated binding affinity of cMslA Gly12-Trp21 was −11.5 kcal/mol.

## Reporting summary

Further information on research design is available in the Nature Portfolio Reporting Summary linked to this article.

## Data availability

Crystal structure data for MslH have been deposited in the Protein Data Bank under accession codes 8GQ9 (MslH:apo), 8ITG (MslH:MslA Trp21G), 8GQA (MslH:MslAΔTrp21), 8GQB (MslH Asp11A:apo), and 8ITH (MslH His295N:apo). The LC-MS, SDS-PAGE, and size-exclusion chromatography data generated in this study are provided in the Source Data file. Any additional data required will be made available upon request. Source data are provided with this paper.

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

## Acknowledgements

This study was supported in part by a Grant-in-Aid for Research on Innovative Areas from MEXT, Japan (JSPS KAKENHI Grant Number JP16H06452 to T.D.), and Grants-in-Aid for Scientific Research from JSPS (JP22H02777 to H.M., JP18H03937 and JP22H04976 to T.D., JP18K05449 and JP22H05130 to Y.O., and JP22K15303 to Y.N.). We thank the Photon Factory staff for beam time allocation and support.

## Author contributions

Y.N., M.M., and M.T. performed screening of crystallizations. Y.N. opti-mized the crystallization condition, and solved and refined the MslH crystal structures. Y.N. performed crystallizations and solved and refined the MslH crystal structures. A.K. and Y.O. performed MslH assays. A.K. and H.M. performed mutagenesis. All authors analyzed data. Y.N., Y.O., T.D. and H.M. wrote the manuscript.

## Competing interests

The authors declare no competing interests.
