## [Peer Review File · Nature Communications]

REVIEWER COMMENTS

Reviewer #1 (Remarks to the Author):

This manuscript describes the structure-based characterization of MslH, an aromatic amino acid isomerase involved in the biosynthesis of MS-271, a known lasso peptide. Previous work from the authors identified MslH as being responsible for installation of D-Trp in MS-271 (Chem Sci 2021, 12, 2567), and they demonstrated the enzyme accepts a linear (pre-lasso) substrate and epimerizes Trp (native), Phe, and Tyr at the C-terminus. In the present work, the authors continue (and correct some) their earlier work on this enzyme. In summary, the work is interesting from an enzymological perspective since it demonstrates a new type of epimerase, but I believe this work, even after revision, fails to meet the sense of urgency and impact that is associated with research published in Nature Communications. For this reason, I suggest that the manuscript be rejected and resubmitted to a more specialized journal after the authors consider the feedback provided below.

1. Throughout the manuscript, the authors state that the metal dependence of MslH was surprising or unexpected. This is based off a faulty assumption from their 2021 paper that EDTA would be capable of stripping all metals from any protein (it is well known that it cannot) and that addition of metals would enhance the activity of the enzyme if they were metal dependent (which is only true for labile metal binding proteins). In fact, the authors should have expected metal dependence. Had they examined their protein using HHPred or InterPro, they would know all other homologs are metal-dependent. And, sequence analysis clearly shows conservation of metal-binding motifs in the protein.
2. It is perplexing that the authors declare they have discovered a new acid/base catalytic mechanism for their epimerase, but they have only performed X-ray crystallography, tested a few variants by a highly qualitative endpoint assay and perform limited modeling (since they could not obtain a structure were the actual Trp that undergoes modification was in the crystal). There are no kinetic assays, there are no pH-rate profiles, there are no KIE studies, or other experiments to substantiate the mechanism they propose.
3. They authors do not perform a rigorous analysis of the mutations they install. There are no data to show that the lack of activity could arise from protein misfolding. There are no data to show if the substrate peptide binds less tightly (K_d or K_m would be acceptable) which would allow for one to declare the residue is actually catalytic (slower k_{cat}). Without these types of biochemical kinetic assays, the authors cannot make the basic conclusions about the role of any of the mutated residues let alone have faith in the proposal shown in Fig6.
4. The authors do not verify the accuracy of their modeling by mutation of the speculative Trp binding site. I would think, at a minimum, k_{cat} and K_m parameters would be needed for L64, I319, Y345, L348, L355, and F356.
5. It is confusing that the authors start the results section with delta-Trp21 with no mention until many pages later that they could not obtain a structure of the actual substrate. It really makes the manuscript more difficult to follow, as it is mystery why they would want to remove the most important portion of the peptide.
6. There are no experiments that evaluate the rates of the forward and reverse epimerization reaction. Is L-Trp ever found in MS-271? How is the D-Trp selected for? Does the enzyme synthesize D-Trp faster than L-Trp (reverse epimerization)?
7. Another major issue with this paper is that the authors imply that an acid/base mechanism for epimerases is a new discovery. This is not true, and a famous example is from Perry Frey's lab on UDP-Gal4-epimerase. There are many other known epimerases that use acid-base catalysis. The authors should be citing literature precedent more generously, and they should clarify that their epimerase is only "new" in the realm of RiPPs and/or its relationship to calcineurin and metal dependence.
8. Dissociation constants are needed to back up the claims made in the penultimate paragraph, which refer to which parts of MslA bind to the enzyme and how stable the interaction is.

Minor points:

1. There are more recent and more comprehensive reviews on the mode of action of RiPPs than what the authors are currently citing.
2. Quite a number of typos in the manuscript: MS-27 instead of MS-271, *S. griseorubiginosus* instead of *griseorubiginosus*, MslA is suggested that it catalyzse a reversible epimerization instead of MslH,

etc.

3. If there is an equal mol percent of Ca and Mn in MslH, how are the authors so certain that Ca is the native metal? The rigor of the explanation through crystallography is unconvincing. The authors need to prepare a Ca-only and Mn-only form of the enzyme and test activity.

4. The authors imply that the lasso cyclization step that follows epimerization is a "tailoring" step, which is a misuse of the term. That is the class-defining, primary modification of lasso peptides. Tailoring is reserved for secondary (non-class-defining) modifications.

5. The penultimate paragraph, especially the last few sentences, are confusing. Please rephrase/clarify.

6. In the last paragraph, the authors suggest that MslH can accept non-MslA substrates, which is true based on their earlier paper. However, the caveats are not discussed, which will severely limit this approach. A full substrate scope analysis has yet to be performed, so caution is warranted.

Reviewer #2 (Remarks to the Author):

The manuscript "Structure of lasso peptide epimerase MslH reveals metal-dependent acid/base catalytic mechanism" by Nakashima and co-workers describes three crystal structures of the unusual amino-acid epimerase MslH, including one complex structure with (truncated) substrate peptide. The structure is novel, and the authors use their structural data in combination with new and published biochemistry as well as docking to propose an intriguing mechanism. It is a very exciting piece of work that allows us to understand yet another enzymatic way to epimerize amino acids.

In my opinion the work reported in this manuscript is of sufficient novelty and broad interest to warrant publication in Nature Communications once the following (mostly) minor issues have been addressed:

Main points:

1. I would like to encourage the authors to address the metal question more fully. Can the metal ions that are bound be removed by extensive dialysis in EDTA? Does protein expression in M9 minimal medium remove the Mn that is detected by ICP? And is the enzyme still active in its (almost) all Ca-bound state that could be obtained that way? That should alleviate any concerns one might have regarding a small Mn-containing fraction of enzyme that is the actually active form, which would also be difficult to detect in refinement at the obtained resolutions.

2. The authors used bacterially expressed precursor peptide truncated at the C-terminus (W missing). I appreciate that they tried to obtain a complex structure with full-length substrate peptide and were unsuccessful. Given the short length of the precursor peptide (just 21 amino acids), have the authors considered using a synthetic, C-terminal amide version of the peptide substrate? Is the enzyme still active with such a substrate? This would also give valuable data for the mechanistic proposal.

3. Mechanism: The mechanism appears to be completely reasonable and in a way remarkably similar to that of BotH, although in the case of MslH it is a metal ion rather than a heterocycle that stabilises the transition state. It would be good to probe the proposed mechanism experimentally, particularly the fate of the Ca proton. Using enzymatically produced, W-Ca deuterated substrate for an enzyme assay in regular water should allow the authors to determine if the proton (or in this case deuterium) is retained in the enzyme during catalysis.

Minor points:

P7, 146 two beta sheets – two beta strands

P7, 159 disappearance – never observed, so does not disappear

P9, 190 metal molecule – metal ion

Reviewer #3 (Remarks to the Author):

Nakashima et al report on the structure and mechanism of MslH, a RiPP epimerase that catalyzes epimerization of the C-terminal Trp residue in the biosynthesis of the lasso peptide MS-271. In order to understand the mechanism of the epimerization, crystal structures were obtained for both MslH:apo and the MslH:MslA Δ W21 complex. The overall structure is most closely related to YmdB phosphodiesterase although noticeable differences were observed outside the calcineurin-like fold. Analysis of the active site residues allowed a six-coordination motif to be proposed. In contrast to a previous report it is now proposed that MslH is Ca(II)-dependent, and a corresponding proposal for the mechanism is presented. The proposed mechanism is acid/base chemistry catalyzed by two pairs of His/Asp catalytic dyads on opposing sides of the Trp.

Overall this interesting paper reports mechanistic insight into a relevant posttranslational modification, peptide epimerisation. There are novel aspects of the enzyme structure and mechanistic proposal, and this paper expands the breadth of knowledge for peptide epimerases. The manuscript is sufficiently detailed and the supporting information is sound. There are two weak points of the manuscript that should be addressed before acceptance.

1. The inability to inactivate the enzyme makes it difficult to say whether the enzyme is Ca(II)-dependent. The data more or less supports that Ca(II) binds during the E. coli expression but it cannot be removed after purification. Could the authors try a chelator cocktail or other chelating agents to try and remove the Ca(II) to show the enzyme is inactive. Alternatively, could the protein be denatured and refolded?
2. The lack of substrate containing the C-terminal Trp residue within the active site. Is it possible to try incubation with variants containing alternate amino acids (Phe, Tyr, or others) at the C-term as these seem to be accepted by the enzyme.

May 4, 2023

NCOMMS-22-52652-T

Title: Structure of lasso peptide epimerase MslH reveals metal-dependent acid/base catalytic mechanism

Responses to Reviewer 1's comments:

1. Throughout the manuscript, the authors state that the metal dependence of MslH was surprising or unexpected. This is based off a faulty assumption from their 2021 paper that EDTA would be capable of stripping all metals from any protein (it is well known that it cannot) and that addition of metals would enhance the activity of the enzyme if they were metal dependent (which is only true for labile metal binding proteins). In fact, the authors should have expected metal dependence. Had they examined their protein using HHPred or InterPro, they would know all other homologs are metal-dependent. And, sequence analysis clearly shows conservation of metal-binding motifs in the protein.

Re: In the previous study, we mentioned that MslH could form a calcineurin-like fold, the typical conformation classically categorized into phosphoesterase domains in metal-dependent phosphodiesterases, such as protein (serine/threonine) phosphatases and 5'-nucleotidases, based on a Phyre2 analysis. Furthermore, the subsequent amino acid sequence alignment suggested that several amino acid residues involved in metal coordination in metallo-dependent phosphatases are conserved in MslH. However, we did not include this information in the current manuscript, considering that no epimerase with calcineurin-like folds has yet been discovered, and the previous EDTA addition experiment led to the incorrect interpretation that MslH was not metal-dependent in our previous paper. According to the reviewer's thoughtful suggestions, we provided the requested information in the "INTRODUCTION" section of the revised manuscript. We also deleted descriptions that could lead to the surprising impression of the metal dependence of MslH throughout the revised manuscript, to avoid misleading the readers, together with the previous metal additive experiment from the "DISCUSSION" section.

2. It is perplexing that the authors declare they have discovered a new acid/base catalytic mechanism for their epimerase, but they have only performed X-ray crystallography, tested a few variants by a highly qualitative endpoint assay and perform limited modeling (since they could not obtain a structure were the actual Trp that undergoes modification was in the crystal). There are no kinetic assays, there are no pH-rate profiles, there are no KIE studies, or other experiments to substantiate the mechanism they propose.

Re: As mentioned by the reviewer, the crystal structure of MslH complexed with full MslA and enzymological data will be required to support our proposed acid/base catalytic mechanism of MslH. However, analyses of the enzymological data such as kinetics and the pH-dependence of MslH are impossible, because we would have to employ the multi-step assay method to monitor the reaction, including the purification of the MslA product using SDS-PAGE after the reaction, its hydrolysis, and Marfey's analysis. We also tried the direct monitoring of MslA→*epi*-MslA reactions by a simple LC-MS analysis, but all trials failed. In addition, we attempted to obtain the crystal structure of MslH complexed with full MslA, but were not successful, due to the reaction of full MslA on MslH during the crystallization. Therefore, we newly performed several alternative experiments to further support the

proposed catalytic mechanism. We obtained the complex structure of MslH with the MslA mutant, MslA Trp21G, which has an achiral glycine instead of Trp21. The crystal structure revealed that MslA Gly21 is accommodated in the cleft, in a location and an orientation very similar to those in the cMslA Gly12-Trp21 docking model, including the C-terminal Gly21 in MslA Gly21 and Trp21 in the cMslA Gly12-Trp21 docking model. Meanwhile, we also performed *in vitro* MslH reactions using MslA in D₂O, to support the proposed MslH reverse reaction. The subsequent LC-MS analysis revealed that MslH produced not only MslA with C-terminal deuterated D-Trp, but also deuterated L-Trp. According to the newly obtained MslH complex structure with MslA Trp21G, we significantly modified the “RESULTS” section, especially the 1st paragraph of the “Overall structure of MslH” section and the 1st and 2nd paragraphs of “Active-site architecture of MslH” section. The descriptions of the docking study of full MslA were shifted to the 3rd paragraph of “Active-site architecture of MslH”, with some modifications. In the revised manuscript, we also updated/added Figs. 2, 4, 5, 7 and Supplementary Figs. 3, 8, 9, 13, 15, and 16 with the addition of the experimental procedures in the “METHODS” sections and the detailed crystallographic data in Supplementary Table 1, according to the newly obtained structural results. Our results obtained from the *in vitro* MslH reactions using MslA in D₂O are provided in the 1st paragraph of “Catalytic mechanism of MslH”, and the experimental data are presented in the new Supplementary Fig. 18 with the experimental procedures in the “METHODS” section.

3. The authors do not perform a rigorous analysis of the mutations they install. There are no data to show that the lack of activity could arise from protein misfolding. There are no data to show if the substrate peptide binds less tightly (Kd or Km would be acceptable) which would allow for one to declare the residue is actually catalytic (slower kcat). Without these types of biochemical kinetic assays, the authors cannot make the basic conclusions about the role of any of the mutated residues let alone have faith in the proposal shown in Fig6.

Re: As mentioned in our response to Reviewer 1’s comment No. 2, the kinetic parameters cannot be obtained in our assay. Since His88/Asp91, among the four MslH residues (Asp11, His88, Asp91, and His295) we proposed as the catalytic residues, correspond to the YmdB catalytic residues, His68 and Asp71, the loss of the MslH activity by the His88 and Asp91 point mutations would explain the His88/Asp91 functions as catalytic residues. In contrast, the loss of catalytic activity of the His295N and Asp11A variants due to a structural collapse of the metal-binding and substrate-binding sites, caused by a breakdown of the hydrogen bonding network between six-coordination and Asp11/His295 via the w2 water molecule and substrate, cannot be excluded. To further strengthen the proposed mechanism from the structural aspect, we newly obtained the MslH His295N apo structure in addition to the MslH Asp11A: apo structure, which has been previously described. Both crystal structures indicated that the variants retained the same six-coordination motif as that in the wild-type MslH crystal structure, as shown in the new Supplementary Figs. 19 and 20. Significant structural changes were also not observed in the overall structures as compared with those of the wild type MslH structures. These results are described in the 1st and 2nd paragraphs of the “Catalytic mechanism of MslH” section, with the experimental procedures in the “METHODS” sections and the detailed crystallographic data in the new Supplementary Table 2.

4. The authors do not verify the accuracy of their modeling by mutation of the speculative Trp binding site. I would think, at a minimum, kcat and Km parameters would be needed for L64, I319, Y345, L348, L355, and F356.

Re: Please see our response above.

5. It is confusing that the authors start the results section with delta-Trp21 with no mention until many pages later that they could not obtain a structure of the actual substrate. It really makes the manuscript more difficult to follow, as it is mystery why they would want to remove the most important portion of the peptide.

Re: We described the reasons why a complex structure of MslH with the wild-type MslA was not obtained at the beginning of the “RESULTS” section.

6. There are no experiments that evaluate the rates of the forward and reverse epimerization reaction. Is L-Trp ever found in MS-271? How is the D-Trp selected for? Does the enzyme synthesize D-Trp faster than L-Trp (reverse epimerization)?

Re: As mentioned in our response to Reviewer 1’s second comment, the *in vitro* MslH reactions were performed in D₂O, and the results supported the proposed reverse epimerization reaction. Because an MS-271 analog containing L-Trp has never been detected from producers, we proposed that a later stage biosynthetic enzyme strictly selected MslA with the C-terminal D-Trp form, as newly stated in the “DISCUSSION” section.

7. Another major issue with this paper is that the authors imply that an acid/base mechanism for epimerases is a new discovery. This is not true, and a famous example is from Perry Frey’s lab on UDP-Gal4-epimerase. There are many other known epimerases that use acid-base catalysis. The authors should be citing literature precedent more generously, and they should clarify that their epimerase is only “new” in the realm of RiPPs and/or its relationship to calcineurin and metal dependence.

Re: To clarify the new points of MslH as an epimerase, we revised descriptions in the 1st paragraph of the “DISCUSSION” section as follows: “To the best of our knowledge, this type of epimerase has never been found in the realm of RiPPs, and this is the first report to demonstrate the relationship to calcineurin-like fold and epimerization.” In addition, UDP-Gal4-epimerase was also mentioned as the example of an “acid/base” epimerase in the last paragraph of the “Catalytic mechanism of MslH” section.

8. Dissociation constants are needed to back up the claims made in the penultimate paragraph, which refer to which parts of MslA bind to the enzyme and how stable the interaction is.

Re: Since the binding mode of the MslA is not the primary discussion in this manuscript, the descriptions pointed out by Reviewer 1 were deleted. Alternatively, we provided descriptions as the last sentences in the “DISCUSSION” section, as follows: “As discussed above, our X-ray crystallographic analysis has provided some clarification of the binding mode of the C-terminal MslA core peptide in the MslH structure. Our observations will facilitate the future production of non-natural lasso peptides with C-terminal D-amino acids, by the enzymatic engineering of the amino acid residues involved in the substrate recognition by MslH.”

Minor points:

1. There are more recent and more comprehensive reviews on the mode of action of RiPPs than what the authors are currently citing.

Re: “Ongpipattanakul, C., Desormeaux, E. K., DiCaprio, A., van der Donk, W. A., Mitchell, D. A. & Nair, S. K. Mechanism of action of ribosomally synthesized and post-translationally modified peptides. *Chem. Rev.* 122, 14722-14814 (2022).” is cited as reference 6 in the revised manuscript.

2. Quite a number of typos in the manuscript: MS-27 instead of MS-271, *S. griseorubiginosus* instead of *S. griseorubiginosus*, MslA is suggested that it catalyzse a reversible epimerization instead of MslH, etc.

Re: The typos, including those mentioned by Reviewer 1, were corrected throughout the manuscript.

3. If there is an equal mol percent of Ca and Mn in MslH, how are the authors so certain that Ca is the native metal? The rigor of the explanation through crystallography is unconvincing. The authors need to prepare a Ca-only and Mn-only form of the enzyme and test activity.

Re: According to the comments by reviewers 2 and 3, we expressed MslH in *E. coli* cultured in M9 medium containing only NaCl, MgSO₄, and CaCl₂, which are essential metals for *E. coli* growth. The *in vitro* enzymatic reaction indicated that the catalytic activity was maintained in MslH expressed in M9 medium, as in the case of the recombinant MslH expressed in Luria-Bertani (LB) medium. We provided this result in the 2nd paragraph, together with the new Supplementary Fig. 17 and experimental procedures in the “METHODS” section.

4. The authors imply that the lasso cyclization step that follows epimerization is a “tailoring” step, which is a misuse of the term. That is the class-defining, primary modification of lasso peptides. Tailoring is reserved for secondary (non-class-defining) modifications.

Re: The sentences were revised as follows: “Finally, MslH releases *epi*-MslA as the end product, which is subsequently subjected to the lasso cyclization steps, and recovers the initial state of His88-Asp91 by proton transfer starting from His295 to Asp91, via the Asp11-w2-Ca(II)-w1-His88 network observed in the crystal structure.”

5. The penultimate paragraph, especially the last few sentences, are confusing. Please rephrase/clarify.

Re: The sentences were rephrased. Please also see the response to Reviewer 1’s comment No. 8.

6. In the last paragraph, the authors suggest that MslH can accept non-MslA substrates, which is true based on their earlier paper. However, the caveats are not discussed, which will severely limit this approach. A full substrate scope analysis has yet to be performed, so caution is warranted.

Re: The sentences were rephrased. Please also see the response to Reviewer 1’s comment No. 8.

Responses to Reviewer 2’s comments:

1. I would like to encourage the authors to address the metal question more fully. Can the metal ions that are bound be removed by extensive dialysis in EDTA? Does protein expression in M9 minimal medium remove the Mn that is detected by ICP? And is the enzyme still active in its (almost) all Ca-bound state that could be obtained that way? That should alleviate any concerns one might have regarding a small Mn-containing fraction of enzyme that is the actually active form, which would also be difficult to detect in refinement at the obtained resolutions.

Re: According to the reviewer’s comment, we expressed MslH in *E. coli* cultured in M9 medium containing only NaCl, MgSO₄, and CaCl₂, which are essential metals for *E. coli* growth. The *in vitro* enzymatic reaction indicated that the catalytic activity was maintained in MslH expressed in M9 medium, as in the case of the recombinant MslH expressed in Luria-Bertani (LB) medium. We provided this result in the 2nd paragraph and the experimental data as the new Supplementary Fig. 17 and experimental procedures in the “METHODS” section.

2. “The authors used bacterially expressed precursor peptide truncated at the C-terminus (W missing). I appreciate that they tried to obtain a complex structure with full-length substrate peptide and were unsuccessful. Given the short length of the precursor peptide (just 21 amino

acids), have the authors considered using a synthetic, C-terminal amide version of the peptide substrate? Is the enzyme still active with such a substrate? This would also give valuable data for the mechanistic proposal.

Re: The synthesis of the MslA leader peptide (21 residues) has been attempted previously, but was not obtained due to its instability. On the other hand, we newly obtained a complex crystal structure of MslH with MslA Trp21G. The structure revealed that the oxygen atom of the C-terminal carboxyl group of MslA Gly21 is located near the metal ion, and that MslH forms hydrogen bonds with the amide bond between MslA Phe20 and Gly21. This is in good agreement with the results observed in the cMslA Gly12-Trp21 docking model, and supports the proposed mechanism. According to the newly obtained MslH complex structure with MslA Trp21G, we significantly modified the “RESULTS” section, especially the 1st paragraph of the “Overall structure of MslH” section and the 1st and 2nd paragraphs of the “Active-site architecture of MslH” section. The descriptions of the docking study of full MslA were shifted to the 3rd paragraph of “Active-site architecture of MslH”, with some modifications. In the revised manuscript, we also updated/added Figs. 2, 4, 5, 7 and Supplementary Figs. 3, 8, 9, 13, 15, and 16 with the addition of the experimental procedures in the “METHODS” section and the detailed crystallographic data in Supplementary Table 1, according to the newly obtained structural results.

3. Mechanism: The mechanism appears to be completely reasonable and in a way remarkably similar to that of BotH, although in the case of MslH it is a metal ion rather than a heterocycle that stabilises the transition state. It would be good to probe the proposed mechanism experimentally, particularly the fate of the Ca proton. Using enzymatically produced, W-Ca deuterated substrate for an enzyme assay in regular water should allow the authors to determine if the proton (or in this case deuterium) is retained in the enzyme during catalysis.

Re: *In vitro* MslH reactions using MslA in D₂O were performed to support the proposed MslH reverse reaction. The subsequent LC-MS analysis revealed that MslH produced not only MslA with C-terminal deuterated D-Trp, but also deuterated L-Trp. We described these newly obtained results in the 1st paragraph of “Catalytic mechanism of MslH”, and provided the experimental data as the new Supplementary Fig. 18 with the experimental procedures in the “METHODS” section.

Minor points:

1. P7, 146 two b sheets – two beta strands

Re: It was corrected.

2. P7, 159 disappearance - never observed, so does not disappear

Re: It was corrected.

3. P9, 190 metal molecule – metal ion

Re: It was corrected.

Responses to Reviewer 3's comments:

Major comments:

1. The inability to inactivate the enzyme makes it difficult to say whether the enzyme is Ca(II)-dependent. The data more or less supports that Ca(II) binds during the E. coli expression but it cannot be removed after purification. Could the authors try a chelator cocktail or other chelating agents to try and remove the Ca(II) to show the enzyme is inactive. Alternatively, could the protein be denatured and refolded?

Re: We expressed MslH in *E. coli* cultured in M9 medium containing only NaCl, MgSO₄, and CaCl₂, which are essential metals for *E. coli* growth. The *in vitro* enzymatic reaction indicated that the catalytic activity was maintained in MslH expressed in M9 medium, as in the case of the recombinant MslH expressed in Luria-Bertani (LB) medium. We provided this result in the 2nd paragraph and the experimental data as the new Supplementary Fig. 17 and experimental procedures in the “METHODS” section.

2. The lack of substrate containing the C-terminal Trp residue within the active site. Is it possible to try incubation with variants containing alternate amino acids (Phe, Tyr, or others) at the C-term as these seem to be accepted by the enzyme.

Re: We obtained the complex structure of MslH with the MslA mutant, MslA Trp21G, which has an achiral glycine instead of Trp21. The structure revealed that the oxygen atom of the C-terminal carboxyl group of MslA Gly21 is located near the metal ion, and that MslH forms hydrogen bonds with the amide bond between MslA Phe20 and Gly21. This is in good agreement with the results observed in the cMslA Gly12-Trp21 docking model, and support the proposed mechanism. According to the newly obtained MslH complex structure with MslA Trp21G, we significantly modified the “RESULTS” section, especially the 1st paragraph of “Overall structure of MslH” section and the 1st and 2nd paragraphs of “Active-site architecture of MslH” section. The descriptions of the docking study of full MslA were shifted to the 3rd paragraph of “Active-site architecture of MslH”, with some modifications. In the revised manuscript, we also updated/added Figs. 2, 4, 5, 7 and Supplementary Figs. 3, 8, 9, 13, 15, and 16 with the addition of the experimental procedures in the “METHODS” section and the detailed crystallographic data in the Supplementary Table 1, according to the newly obtained structural results.

Additional revisions by authors:

1. The equal contribution as the first author was provided to Atsushi Kawakami, the 2nd author.

REVIEWERS' COMMENTS

Reviewer #1 (Remarks to the Author):

The authors have put forth a very strong and convincing response to the prior review, including many new data that strengthen the manuscript substantially. I am of the opinion this version of the manuscript should be published in Nature Communications.

Reviewer #2 (Remarks to the Author):

The authors have gone to great lengths to revise their manuscript. Especially the complex crystal structure of MslH with the W21G mutant, which shows how the carboxy terminus is coordinated by the metal ion, is very convincing with regards to the proposed mechanism.

The identity of the metal ion appears to be settled (Ca^{2+}), and the only other experiment I could think of would be an XRF scan of the crystals from protein obtained from M9 medium.

With regards to more elaborative experiments suggested to probe the mechanism: I understand the desire to probe this more fully, but it appears to me to be beyond the scope of this manuscript. An amino acid epimerase, which provides a substrate pool for a downstream enzyme, that then provides stereochemical resolution for the pathway has been reported for RiPPs before, but NOT for lasso peptides, and certainly not for a calcineurin-like fold.

Given the novelty of the findings and their potential to inform future biosynthetic studies, as well as genome mining, I am of the opinion that the manuscript is very suitable for publication in Nat Comm. Given the revisions provided by the authors I have no further revision requests.

Reviewer #3 (Remarks to the Author):

This reviewer is satisfied with the additional experiments and revised manuscript. It is now acceptable for publication.